# Full-Length Transcriptomes and Sex-Based Differentially Expressed Genes in the Brain and Ganglia of Giant River Prawn *Macrobrachium rosenbergii*

**DOI:** 10.3390/biom13030460

**Published:** 2023-03-02

**Authors:** Dong Liu, Zhenzhen Hong, Lang Gui, Li Zhao, Yude Wang, Shengming Sun, Mingyou Li

**Affiliations:** 1Key Laboratory of Integrated Rice-Fish Farming, Ministry of Agriculture and Rural Affairs, Shanghai Ocean University, Shanghai 201306, China; 2Key Laboratory of Exploration and Utilization of Aquatic Genetic Resources, Ministry of Education, Shanghai Ocean University, Shanghai 201306, China; 3Shanghai Universities Key Laboratory of Marine Animal Taxonomy and Evolution, Shanghai Ocean University, Shanghai 201306, China; 4State Key Laboratory of Developmental Biology of Freshwater Fish, College of Life Sciences, Hunan Normal University, Changsha 410081, China

**Keywords:** prawn, RNA-seq, gene, signal pathway, LncRNA

## Abstract

*Macrobrachium rosenbergii* is an important aquaculture prawn that exhibits sexual dimorphism in growth, with males growing much faster than females. However, the mechanisms controlling these complex traits are not well understood. The nervous system plays an important role in regulating life functions. In the present work, we applied PacBio RNA-seq to obtain and characterize the full-length transcriptomes of the brains and thoracic ganglia of female and male prawns, and we performed comparative transcriptome analysis of female and male prawns. A total of 159.1-Gb of subreads were obtained with an average length of 2175 bp and 93.2% completeness. A total of 84,627 high-quality unigenes were generated and annotated with functional databases. A total of 6367 transcript factors and 6287 LncRNAs were predicted. In total, 5287 and 6211 significantly differentially expressed genes (DEGs) were found in the brain and thoracic ganglion, respectively, and confirmed by qRT-PCR. Of the 435 genes associated with protein processing pathways in the endoplasmic reticula, 42 DEGs were detected, and 21/26 DEGs with upregulated expression in the male brain/thoracic ganglion. The DEGs in this pathway are regulated by multiple LncRNAs in polypeptide folding and misfolded protein degradation in the different organs and sexes of the prawn. Our results provide novel theories and insights for studying the nervous system, sexual control, and growth dimorphism.

## 1. Introduction

*Macrobrachium rosenbergii*, the giant river prawn, lives in fresh water in northern Australia and Southeast Asia. This species is the largest *Macrobrachium* species, with complex traits such as an omnivorous diet, faster growth, and a short reproductive cycle. It has become a popular species in prawn farms and is commercially cultured in many countries and regions [1]. *M. rosenbergii* was first introduced to China (from Japan) in 1976 and was quickly cultivated on a large scale. Its cultivation has maintained an increasing trend and reached 139.6 thousand tons of live weight in aquaculture in 2020, accounting for 47.48% of the total global production (294 thousand tons) [2,3].

A striking feature of the giant river prawn is that males grow faster and are significantly larger than adult females. In addition, the second pair of chelipeds of males are larger and thicker than those of females, exhibiting sexual dimorphism [4]. Genetic sex determination in *M. rosenbergii* follows the ZW mode, and WW females can sex reverse into functional males via androgenic gland cell transplantation [5]. The physiological molting of giant river prawns is a fundamental process of the post-larvae prawn, through which they realize metamorphosis, growth, and development. Successful molting is regulated by both the nervous and neuroendocrine systems, exhibiting an initial rise and a coordinated decline in the circulating concentration of molt hormone and molt-inhibiting hormone [6]. Genes in the ecdysone signaling pathway underlying hormonal regulation have been identified as involved in molting and epidermis reconstruction. These include chitin binding proteins, crustacean hyperglycemic hormone, calcification-related cuticular proteins, and chitinase in penaeid white shrimp (*Litopenaeus vannamei*) [7]. Based on family-selected stocks of *Litopenaeus vannamei*, a comparison of transcriptomes of higher and lower growth revealed genes potentially involved in superior growth performance [8]. However, little attention has been paid to regulating sexual dimorphism in growth by the nervous system, which could be helpful in understanding the nervous regulation of life processes [9].

In evolutionary biology, the nervous system tends to concentrate and progress from a scattered and superficial nervous net to a complex, centralized nervous system. The nervous system allows animals to regulate body functions by generating, modulating, and transmitting information [10]. The brain of *M. rosenbergii* is located in a mass of spongy tissue within the base of its eyestalks. It comprises the protocerebrum, deutocerebrum, tritocerebrum, nerve roots, commissures, and clusters of cell bodies working in concert. Its rope-ladder-like nervous system is composed of cerebral, thoracic, and abdominal ganglia [11]. Extracts from the thoracic ganglia stimulate oocyte development in the giant river prawn [12], which acts indirectly on the gonads by triggering the release of a putative gonad-stimulating factor from the thoracic ganglion [13]. Secretions from decapod crustaceans’ eyestalk, brain, and thoracic ganglia have a stimulatory effect on ovarian maturation [14]. The eyestalk’s regulation of ovarian maturation has been well-studied by Illumina sequencing in the Pacific white shrimp, *Litopenaeus vannamei* [15]. Due to a lack of completely related gene analysis, the possible molecular mechanisms by which the brain and thoracic ganglion regulate gonad development and growth remain unclear. RNAseq-based full-length transcriptome sequencing methods are capable of sequencing transcripts from end to end at a single molecule level and can be used to annotate transcriptome structures in a variety of organisms [16]. However, comparative studies of the transcriptomes for the brains and thoracic ganglia between sexes of the giant river prawns have received little attention outside miRNA [17].

To investigate and better understand sex and growth differences through regulation of the brain and thoracic ganglion in *M. rosenbergii*, we analyzed the full-length transcriptomic characterization of the brains and thoracic ganglia of female and male prawns using SMART single molecule sequencing and compared gene expression differences between the sexes. This allowed us to discover that differentially expressed genes were significantly enriched in the endoplasmic reticulum pathway’s protein processing, a unique protein-synthesizing mechanism that plays a crucial role in efficient neural communication in the giant river prawn. To our knowledge, this is the first full-length transcriptome-wide gene expression analysis of the brain and thoracic ganglion of *M. rosenbergii*. This study will serve as an essential resource for further studies of growth regulation mechanisms and is of particular interest in molecular breeding to target growth genes.

## 2. Materials and Methods

### 2.1. Tissue Material

Three male and three female mature prawns in good health were randomly selected as experimental specimens from Huzhou Fengsheng Bay Farm, Zhejiang province, China, and subjected to an ice bath for immobilization. The brain and thoracic ganglion tissues of each prawn (approximately 20 g) were rapidly excised and flash-frozen in liquid nitrogen. The dissected tissues were stored at −80 °C until the total RNA was extracted. Animal experiments were approved by the Animal Ethics Committee of Shanghai Ocean University.

### 2.2. RNA Extraction and Compete Transcript Sequencing

To generate one SMART library, total RNA was extracted from the mixed tissues of three male and three female individuals using the Phygene Total RNA Isolation Kit (Phygene, Fuzhou, China) and following the manufacturer’s instructions. RNA purity was determined first by agarose gel electrophoresis, followed by the use of a NanoDrop instrument (IMPLEN, Westlake Village, CA, USA). RNA quantity was measured using an Agilent 2100 Bioanalyzer (Agilent Technologies, Santa Clara, CA, USA). The poly (A) mRNA was isolated from the total RNA via Oligo (dT), and the mRNA was reversed into full-length first-strand cDNA using a SMARTer PCR cDNA Synthesis Kit. The synthesized cDNA was enriched using PCR amplification. Fragment screening of partial cDNAs was performed by the BluePippin Size Selection System, and the 5–10 kb size optional fragments were selected and subsequently enriched using PCR amplification. The amplicators were used to construct a SMART library, subsequently sequenced on the PacBio RS II sequencing platform (Pacific Biosciences, Menlo Park, CA, USA), and final complete reads were generated.

### 2.3. Data Processing and Full-Length Transcript Functional Annotation

The raw data (subreads) from SMART sequencing were filtered and corrected to obtain circular consensus sequences, in which the adaptors, barcodes, polyA, and chimera were eliminated and then polished using isoseq3 software [18] to obtain isoform sequences. After removing low-quality isoform sequences, based on min passes = 2 and min predicted accuracy = 99%, the high-quality isoform sequences were obtained and clustered into a unigene sequence via CD-HIT [19] with identification of 98% model. The completeness of the unigene sequence was assessed by BUSCO with the arthropoda_odb9 database [20] and annotated functionally using diamond software, with an e value of e < 1e − 5 based on six different databases: non-redundant sequences (NR), eukaryotic ortholog groups (KOG), Gene Ontology (GO), Swissprot, Evolutionary Genealogy of Genes Non-supervised Orthologous Groups (eggNOG), and Kyoto Encyclopedia of Genes and Genomes (KEGG) databases [21,22]. The protein families were assigned by the HMMER 3.1 package (http://hmmer.org/download.html, accessed on 1 July 2021) with the Pfam database [23]. Venn diagram of the intersection genes annotated in various databases was constructed by TBtools [24].

### 2.4. Identification of TF, CDS, and lncRNAs

The transcription factors (TF) were identified using blast and compared against the AnimalTFDB database [25]. Coding sequences (CDS) of the unigenes were annotated through blast against the NR, Swiss-Prot, and KOG databases. The long non-coding RNAs (lncRNAs) were predicted from transcripts without coding potential using Coding Potential Calculator 2 [26], Coding Non-coding Index [27], Pfam [23], and PLEK [28], with minimum length of 200 bp and minimum ORF of 300 bp as the cut-off criterion.

### 2.5. Differential Expression Analysis via Illumina cDNA Library Sequencing

To prepare the Illumina library, total RNA was extracted from the brain and ganglion tissues of three male and three female individuals as described above. Three independent biological replicates were performed for each based on sex tissues. The mRNAs were enriched with Oligo (dT) mRNA magnetic beads following the manufacturer’s instructions (TruSeq Stranded mRNA Library Prep Kit, Illumina, San Diego, CA, USA). The cDNA libraries were synthesized using mRNA fragments as templates. The quality of the cDNA Illumina libraries was checked using an Agilent 2100 Bioanalyzer (Agilent Technologies, Santa Clara, CA, USA), and 150 bp paired-end reads were generated by Illumina HiSeq Ten platform. The unigene sequences generated by SMART sequencing were used as a reference. The clean reads obtained from Illumina sequencing were aligned to the reference using bowtie2 [29] to analyze gene expression counts. The fragments per kilobase of transcript per million mapped reads (FPKM) values for the gene expression level were calculated using eXpress [30]. Based on gene expression counts obtained from each sample, differentially expressed genes were identified using DESeq [31]. The thresholds for significant differential expression were set as a *p*-value < 0.05 and absolute of log_2_(fold change) > 2. Significance was tested using hypergeometric distribution. Finally, differentially expressed genes were used for GO and KEGG enrichment analysis.

### 2.6. Differentially Expressed Gene Validation via qRT-PCR

Ten differentially expressed genes (DEGs) were selected to verify their differential expression. Among these genes, three showed up-/down-regulated expression in the brain, and three showed up-/down-regulated expression in the ganglion. Of them, one up-regulated expression in both the brain and ganglion, and one down-regulated expression in the brain while up-regulating expression in the ganglion. The qRT-PCR primers were designed using Primer 6 and are listed in Table 1. The *actin* gene was used as a control. The qRT-PCR was carried out on a real-time PCR system (Eppendorf, Hamburg, Germany). The reacted mixture consisted of 1 μL cDNA (60 ng/μL), 10 μL Bestar SYBR Green qPCR Mastermix (DBI, Bioscience Inc., Hamburg, Germany), 1 μL of each primer, and 7 μL mill i-Q water. Reactions were performed at 95 °C for 30 s; 30 cycles of 95 °C for 5 s, 60 °C for 30 s, and 72 °C for 60 s. The qRT-PCR results were obtained from three biological replicates, and the gene relative expression levels were calculated using the 2−ΔΔCt method.

### 2.7. Construction of LncRNA and Gene Co-Expression Networks

A network of LncRNA gene co-expressions was constructed using the Pearson correlation coefficient between the differentially expressed LncRNAs and differentially expressed mRNAs using the FPKM values of four tissue samples via Illumina sequencing. The threshold for positive correlation was set to PCC > 0.95 and *p*-value < 0.05. The regulatory network of LncRNAs and mRNAs was established via Cytoscape [32], and the ZFLNC database [33] was used to conduct a conservative analysis of lncRNAs.

## 3. Results

### 3.1. Full-Length Transcription Sequences

Brain and thoracic ganglion samples of sexually mature giant river prawns were used for single-molecular real-time (SMART) sequencing. After removing adaptor sequences and low-quality sequences, a total of 73,146,314 subreads (159.1 Gb) were obtained, and the average length of the subread was 2175 bp. Subread sequences were subjected to self-correction (min passes = 2, min predicted accuracy = 0.8) to produce 1,725,793 circular consensus sequence reads (CCS), and the CCS sequences were clustered into 1,509,283 full-length non-chimerics (FLNC). The FLNCs were polished to obtain 99,728 high-quality isoform numbers (288 Mb), with an average length of 2891 bp. Finally, 84,627 unigene sequences (246 Mb) with an average read length of 2913 bp were obtained using high-quality isoform sequence clustering (identity = 98%). A total of 78,559 transcriptions were found between 1 and 10 isoforms. The transcript length extended from 170 bp to 14,287 bp, with an average length of 3061 bp.

### 3.2. Assessment of Unigene Completeness

The completeness and accuracy of the obtained unigenes were assessed via BUSCO with the arthropoda-odb9 database. Compared to 60 species with 1066 available gene sequences in the database, 994 (93.2%) sequences were completely homologous to matches within the BUSCO database (Appendix A). Our SMART data were determined to be of high quality and could be used for subsequent analysis.

### 3.3. Functional Annotation of Unigenes

All 84,627 SMRT sequences from these unigenes were functionally annotated by searching seven databases: NR, Swiss-Prot, KEGG, KOG, eggNOG, GO, and Pfam. A total of 62,612 (73.99%) unigenes were annotated in at least one database. The most was 57,846 (68.35%) unigenes, which were functionally annotated in the NR database. The least was 5192 (6.14%) unigenes, which were functionally annotated in the KEGG database. A total of 33,808 genes were obtained from the intersection of the seven databases (Appendix A). The functional annotations of all 84,627 unigenes are listed in Appendix A.

Homologous species analysis of all unigenes annotated in the NR database showed that *Hyalella azteca* was the species of giant river prawn with the most unigene sets (21,754, 37.6%), followed by *Cryptotermes secundus* (3136, 5.42%), *Procambarus clarkii* (2702, 4.67%), *Zootermopsis nevadensis* (2523, 4.36%), and *M. rosenbergii* (1749, 3.02%) (Appendix A). GO analysis demonstrated that 46,661 unique genes were annotated as being associated with binding in terms of biological process (41,379), cellular component (43,948), and molecular function (41,174). In these three categories, the top three abundant subcategories were: cellular process (79.1%), metabolic process (56.1%), and biological regulation (53.6%) in the biological process; cell (86.1%), cell part (85.9%), and organelle (68.4%) in cellular component; binding (71.8%), catalytic activity (41.7%), and transporter activity (7.9%) in molecular function (Figure 1).

A total of 5192 unigene sequences were annotated by the KEGG data and plotted according to 87 classifications. Genetic information processing was the largest transcript category, including translation, transcription, folding, sorting, and degradation (Figure 2). The first four transcript-related pathways in genetic information processing were spliceosome (505, 9.7%), ribosome (471, 9.1%), protein processing in the endoplasmic reticulum (435, 8.4%), and RNA transport (4335, 8.3%) (Appendix A). Gene function was classified by KOG data, and most were defined specifically in signal transduction mechanisms, posttranslational modification, protein turnover, chaperones, cytoskeleton, intracellular trafficking, secretion, and vesicular transport (Appendix A). A large number of KOG-annotated genes were related to neuron cell commitment and sensation, such as neuropeptide signal transduction (122), innervation activity (50), sensory perception of sound (357), sensory perception of light stimulus (22), and neurotransmitter biosynthesis including gamma-aminobutyric acid (144), dopamine secretion (77), G protein-coupled serotonin receptor (10), and histamine secretion (7) (Appendix A).

### 3.4. Results of TF, CDS, and LncRNAs

Transcription factors (TFs) regulate gene expression during the transcription of DNA into RNA by binding to particular regions of DNA. The unigene sequences of giant river prawn were blasted against AnimalTFDB to predict TFs. Finally, a total of 6367 putative TFs across 58 families were identified from these unigene sequences (Appendix A). The top 30 annotated families are shown in Figure 3. The top 10 species distributions included 945 (14.84%) from *Drosophila melanogaster*, 641 (10.07%) from *Homo sapiens*, and 448 (7.04%) from *Danio rerio* (Appendix A).

Of the 70,536 transcripts, 57,963 CDS were identified by blast against the NR, Swiss-Prot, and KOG databases. In total, 63.4% of these transcript CDSs’ lengths ranged between 201 bp and 1400 bp (Figure 4A). LncRNAs were predicted by CPC2, CNCI, Pfam, and PLEK. Overall, 6287 LncRNAs were identified (Figure 4B). Their lengths varied from 200 bp to 6482 bp. A majority of lncRNA lengths (>71%) ranged between 1000 and 3000 bp, and the average length was 1557 bp, shorter than that of mRNA transcripts (Figure 4C).

### 3.5. Differentially Expressed Genes in Brain and Thoracic Ganglion between Sexes Based on Illumina Sequencing Results

Illumina sequencing was performed on the female and male prawns’ brain and thoracic ganglion tissues with three independent biological replicates. After processing the raw reads, clean reads were obtained. The percentage of valid bases in all reads exceeded 92%, and the Q30 value of all sequences per library was as high as 94% (Appendix A). Using the annotated unigenes as reference sequences, Illumina sequences were mapped to the reference sequences. The mapped rates were 83.92% (male thoracic ganglion) to 90.49% (male brain) for all Illumina sequences in each library. Further, 79.06–85.76% of the Illumina sequences were matched in pair-ended reads (Appendix A).

Volcano plots showed that the sex-based gene expression levels were distinguishable and statistically significant (*p*-value < 0.05 and the absolute value of the log_2_-fold change >2) (Figure 5). Compared with female prawns, a total of 5287 and 6211 significantly differentially expressed genes (DEGs) were displayed in the male prawn brain and thoracic ganglion, respectively. Out of them, 2416 and 3280 DEGs were upregulated, 2871 and 2931 DEGs were down-regulated, and 3575 and 4499 DEGs were exclusively expressed. A total of 1712 DEGs were shared by male and female prawns. GO annotation of DEGs revealed the top up-regulated genes in biological processes, cellular components, molecular function categories, and their involvement in hemolymph coagulation, extracellular region, and lipid transporter activity for the brain (Appendix A), as well as cell surface pattern recognition receptor signaling pathways, 4-aminobutyrate transaminase complex, and its activity for the thoracic ganglion (Appendix A). For the growth-related genes in GO molecular function, the top number of genes was actin binding (GO: 0003779), which was up to 80 up-regulated expression in the male brain (*p*-value 0.0009, and q-value 0.0182), followed by transforming growth factor beta receptor activity (GO: 0005026) with three down-regulated expressions in male thoracic ganglion and two up-regulated expressions in both the male brain and thoracic ganglion (*p*-value < 0.05, and q-value < 0.05). KEGG pathway annotation showed that DEGs were sorted into translation, transcription, protein folding, sorting, and degradation for the brain (Appendix A) and thoracic ganglion (Appendix A).

### 3.6. Validation of Differentially Expressed Genes

To validate the DEGs, ten genes were selected using a qRT-PCR method to determine their relative expression levels. These genes included *Smc6* (structural maintenance of chromosomes protein 6) in ubiquitin protein ligase binding activity, *Iagbp* (insulin-like androgen gland hormone binding protein) in feedback IAG signaling, *Aspm* (abnormal spindle-like microcephaly associated protein homolog) in the functional regulation of neurogenesis and brain growth, *fem1b*, *transformer2*, and *female-lethal d* in sex determination and differentiation, *Ddb1* (DNA damage-binding protein 1) in protein processing-related E3 ubiquitin ligase complex, *Hsp90b1* (heat shock protein 90 beta 1) in stabilizing and folding other proteins, *Gsg2* (ganglioside GM2 activator) in reductions of GM2 gangliosidosis (mutations in this gene result in Alzheimer’s disease in humans), and *alpha 2 macroglobulin* gene. Gene expression results from qRT-PCR were consistent with FPKM values from Illumina sequencing under the same conditions, except for *fem1b* in the brain (Figure 6), suggesting that DEGs from RNA-seq data were reliable.

### 3.7. Gene Expression Analysis in Protein Processing Signaling Pathways

In DEG-enriched pathways, up-regulated expression of male prawn genes associated with protein processing in the endoplasmic reticulum plays an essential role in protein folding and elimination of misfolded protein in the brain (Figure 7A) and thoracic ganglion (Figure 7B). In fact, male prawns showed faster growth than females, and more proteins needed to be assembled. In turn, more misfolded proteins are to be removed to avoid concentrations exceeding the threshold for proteotoxic stress. A total of 435 genes involved in protein processing in the endoplasmic reticulum were identified (Appendix A). Principal component analysis (PCA) separated these genes expressed in the brains and thoracic ganglia of males and females into distinct principal components. The first and second principal components accounted for 40.9% and 36.2% of the variation between the sexes, respectively (Figure 7C). Of the 42 DEGs between the prawn sexes, 21 male genes had up-regulated expression in the brain, and 26 had up-regulated expression in the thoracic ganglion compared with the female genes. A total of 38 DEGs from male prawns and 39 DEGs from female prawns were identified (Figure 7D). These DEGs, shown in a heat map, support the results of the PCA analysis.

The DEGs were involved in protein recognition by luminal chaperones (k09486, 3 genes), polypeptide folding, and correction (k1718, three genes). Most DEGs were involved in the ER-associated degradation pathway (ERAD), a process that begins with the recognition of misfolded proteins, followed by ubiquitination and retrotranslocation to the cytosol for degradation in the proteasome (Figure 8). Protein disulfide-isomerases (PDIs, K09580, ten genes) and Osteosarcoma-9 (OS-9, k10088, one gene) delivered misfolded polypeptide to the ubiquitin ligase complex, composed of ubiquitin-protein ligases such as Dao10 (k10661, three genes), Sel1L (k14026, four genes), and Hsp40 (k09502, five genes) in ERAD. A translocon-associated protein, TRAP (k13251, three genes), showed up-regulated expression in the male brain and thoracic ganglion, which plays a role in accelerating ER degradation of unfolded proteins (Figure 8).

### 3.8. The Network of Differently Expressed lncRNA Gene Co-Expression in ERAD

The collaboration between the differentially expressed (DE-) lncRNAs and the differentially expressed (DE-) mRNAs was examined, and co-expression analysis of the DE-lncRNAs and the corresponding DE-mRNAs based on FPKM values showed 5086 pairs with 26 DE-lncRNAs and 1328 DE-mRNAs in the brain and nerve ganglion. Out of them, 20 DE-lncRNAs and 6 DE-mRNAs in the co-expression network were enriched in the ERDA pathway, with a Pearson correlation coefficient |R| > 0.95 and *p*-value < 0.05 (Figure 9A). The lncRNA DLEU2 family has four elements, and DLEU2_2 & 5 element’s degree was four, suggesting their potential interacting partners of DE-mRNAs, respectively. For the DE-mRNAs, PDIs.2’s degree is equal to 17, the highest interacting partners of DE-lncRNAs, indicating the function regulation of PDIs.2 by multiple DE-lncRNAs in the ERDA pathway. The six DE-mRNAs showed a similar model of expression in the brain (Figure 9B) and ganglion (Figure 9C), indicating that the co-expressed genes in ERDA may play a critical role in the misfolded protein degradation of *M. rosenbergii* which represent the difference in body size between female and male.

## 4. Discussion

Using the PacBio RS II sequencing platform, a 159.1 GB subread base was generated with 73,146,314 subreads and 2175 average reads. In total, 84,627 unigenes were detected in *M. rosenbergii*. All unigene sequences were scored using BUSCO. A total of 93.2% of the unigenes had high homology and complete matches, indicating that our full-length cDNA sequences are a rich resource for further functional genomics analysis in *M. rosenbergii*.

All unigene sequences were functionally annotated using the NR database. In total, 37.6% of the unigenes were aligned with *Hyalella Azteca*, a widely distributed species of amphipod crustacean, followed by decapod crustaceans with lower match rates (4.67% for *Procambarus clarkii* and 3.02% for *M. rosenbergii*). These results suggest that our full-length transcripts provide additional novelty and complexity for functionally important proteins previously unannotated in the *M. rosenbergii* transcriptome. Transcripts annotated using the GO database were enriched in categories associated with biological processes, cellular components, and molecular functions. In addition, they were enriched in subcategories such as metabolic processes, biological regulation, catalytic activity, and transporter activity. According to KEGG annotations, genetic information processing was the dominant enriched transcript group, especially in pathways for spliceosomes, ribosomes, protein processing in the endoplasmic reticulum, and RNA transport. Interpretation of gene function of SMRT transcripts using the KOG database revealed that a large number of genes are functionally involved in neuron cell commitment, neuropeptide signal transduction, and neurotransmitter biosynthesis. This might provide clues for further investigation of brain and nerve ganglia regulation in sexual dimorphism about the growth, development, and morphology of *M. rosenbergii*. Neurosecretory cells in the brain produce various neuropeptides with regulatory effects on biological functions, and allatostatin, crustacean female sex hormone, crustacean hyperglycemic hormone, and eclosion hormone were identified from the transcriptome of Chinese mitten crab brain [34].

Transcription factors are proteins that modulate the transcription of specific genes by binding to DNA, thereby regulating the proliferation, migration, and differentiation of neural tissues [35,36]. In *M. rosenbergii*, 6367 putative transcription factors were identified, and their function requires further investigation. LncRNAs constitute an important layer of regulation of gene expression at the transcriptional or post-transcriptional level during fundamental processes [37]. Transcriptome profiles of LncRNAs in the brain of zebrafish showed significant sex differentiation between male and female individuals [38]. A total of 6287 LncRNAs were obtained from *M. rosenbergii* using four analytical methods. The number is similar to that of Platypus but significantly lower than other mammals [37]. The evolution of LncRNA and their biological function are driven by transposable elements, particularly transposable elements inserted at transcription start sites that control the transcription of LncRNA [39].

According to the Illumina sequences from the brain and nerve ganglion of *M. rosenbergii*, a large number of DEGs were found and annotated using the GO database. Growth function was a prevalent term in the GO database. In the giant freshwater prawn, periodic shedding of the exoskeleton is one of the major physiological processes essential for growth, including molting and muscle development. Genes related to molting, such as molt-inhibiting hormone, crustacean hyperglycemic hormone, ecdysteroid receptor, and retinoic acid X receptor [7], were not found in DEGs in the present study. In contrast, growth-associated gene expressions were observed in DEGs. In particular, *actin* is one of the most abundant intracellular proteins and preferential binding of actin-beta to myosin, which contains actin binding sites. It plays a key role in many essential biological processes for cell adhesion, migration, and contractility in muscle [40]. The increase in myofiber numbers (hyperplasia) is important for body growth in aquatic animals. For example, the sea bream *Sparus aurata*’s muscle hyperplasia contributes significantly to its adult size, while the zebrafish *Brachydanio rerio* shows slight hyperplasia and reaches only a tiny adult size [41]. In the present study, GO enrichment results from 80 and 66 up-regulated genes involved a common actin binding process in the male brain and nerve ganglion, respectively, indicate that the faster growth of males compared to females in the giant freshwater prawn is under the nervous system control. In the DEGs, the insulin-like androgenic gland hormone binding protein (IAGBP) gene showed up-regulated expressions in male brains. It has been demonstrated that the IAGBP, as a binding partner of the insulin-like androgenic gland hormone (IAG), directly increases IAG transcripts, which play an important role in the growth and development of male *M. rosenbergii* [42]. Meanwhile, specific growth factors, such as transforming growth factor beta (TGF-beta) and p38-MAPK binding proteins [43], were up-regulated in the male brain and nerve ganglion. TGF-beta is a central mediator of fibroblast activation. In skeletal muscle, TGF-beta signals by binding to tissue-specific combinations of receptor subtypes, triggering the activation of a muscle hypertrophy program, and promoting muscle growth [44]. In the brain/nerve ganglion, differentially expressed genes between male and female prawns lay the foundation for the further study of gene expression and manipulation in growth and development in giant freshwater prawn.

KEGG pathway analysis showed that DEGs were significantly enriched in protein processing in the endoplasmic reticulum pathway, which not only represents a unique protein-synthesizing mechanism, but also is a crucial factor for efficient neural communications [45]. Transport from the endoplasmic reticulum (ER) to the Golgi is an essential cellular process in the secretory pathway, and selective export of proteins from the ER requires transport signals for efficient recruitment of a transmembrane intracellular lectin ER–Golgi intermediate compartment protein 53 (ERGIC-53) into budding vesicles [46]. Overexpression of exogenous ERGIC-53 has been shown to increase protein-secreted production in batch cultivation [47]. ERGIC-53, which was used as an ER export determinant, was found to be highly expressed only in the male nerve ganglion of giant freshwater prawn, suggesting that it facilitates the secretion of protein, including neuropeptides and hormones that are transported to the Golgi for delivery to their final destinations. Rainbow trout that are transferred from freshwater to seawater experience growth stunting and down-regulation in growth factor IGF-I transcription, revealing endoplasmic reticulum stress to be a key mechanism underlying this growth stunting phenotype [48]. The maintenance of intracellular hormones can be controlled by correctly folded models, and misfolded proteins in the endoplasmic reticulum are eliminated by a highly effective protein degradation process known as ERAD. We found multiple genes involved in terminally misfolded protein elimination in ERAD. Members of two gene families were significantly upregulated in male giant freshwater prawns, Derlin in the brain and SVIP in the nerve ganglion. Derlin-2 is homologous with Derlin-1 and capable of recognizing the misfolded protein. It physically interacts with p97, SEL1L, and SVIP to form a retrotranslocated complex, thereby, using the Derlin-1 channel, the misfolded protein is transported into the proteasome for degradation via ERAD [49,50]. Therefore, it is conceivable that male prawns are more likely than females to remove misfolded proteins, since the male giant freshwater prawns grow faster than the females.

To explain the regulatory mechanism underlying DEGs in protein processing in the endoplasmic reticulum, a LncRNA and mRNA co-expression network analysis was performed to find differentially expressed LncRNAs that regulate transcription of DEGs in the nervous system, such as LncRNAs, DLX6-AS1, ZNFX1-AS1, and HULC, which have been reported to play critical roles in promoting oncogenic phenotypes of cancer cell lines [51,52]. LncRNAs are highly expressed in the nervous system and control gene expression in the developing and adult brain, mediating neural differentiation [53]. However, these conclusions have some limitations due to the comparative analysis of next-generation sequencing data from the tissues. Further in vivo studies will help to understand the role of LncRNAs in the nervous system of the giant freshwater prawn.

## 5. Conclusions

In this study, RNA-seq in *M. Rosenbergii* with SMRT technology was applied to the tissues of the brains and thoracic ganglia from male and female samples. Based on the results of PacBio sequencing, transcript’s functional annotation, transcription factors, and long non-coding RNA (LncRNA) predictions were made. The results provide significant information for transcriptomic characterization that might be useful for understanding the nervous system regulation of growth dimorphism, gene functions related to sexual development, and the molecular-guided breeding programs of *M. Rosenbergii*. A comparative transcriptome analysis of sex-based brain and ganglion tissues was performed to identify the differential expression of some genes, especially those related to RNA transport and protein processing in the endoplasmic reticulum, which are involved in muscle growth and development in the different organs and sexes of the giant river prawn. These findings provide crucial information for understanding sexual dimorphism in the growth of *M. Rosenbergii*.

## Figures and Tables

**Figure 1 biomolecules-13-00460-f001:**
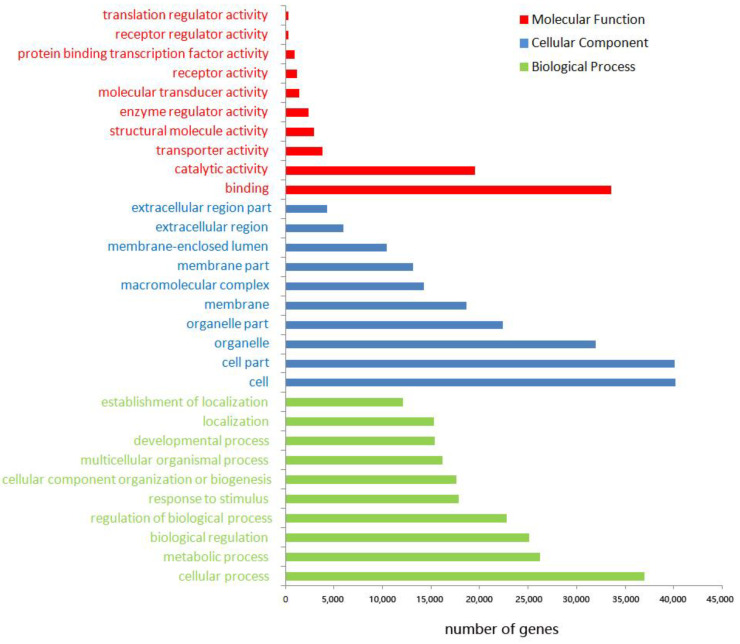
Function annotation of unique transcriptions of *M. rosenbergii* by GO database.

**Figure 2 biomolecules-13-00460-f002:**
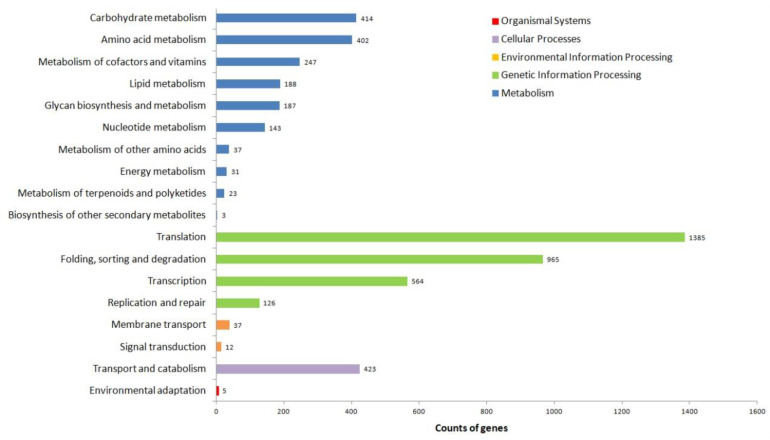
Annotated pathways and numbers of unique transcripts of *M. rosenbergii* by KEGG database.

**Figure 3 biomolecules-13-00460-f003:**
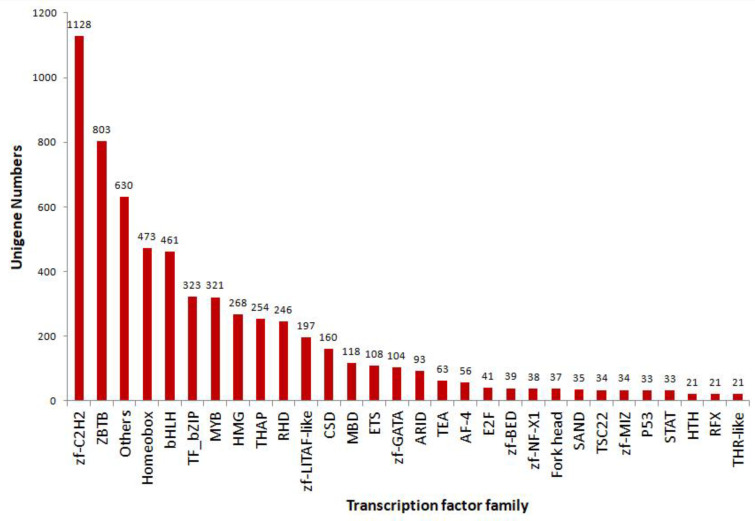
Numbers and families of the top 30 transcription factors in *M. rosenbergii* predicted by comparison against the AnimalTFDB database.

**Figure 4 biomolecules-13-00460-f004:**
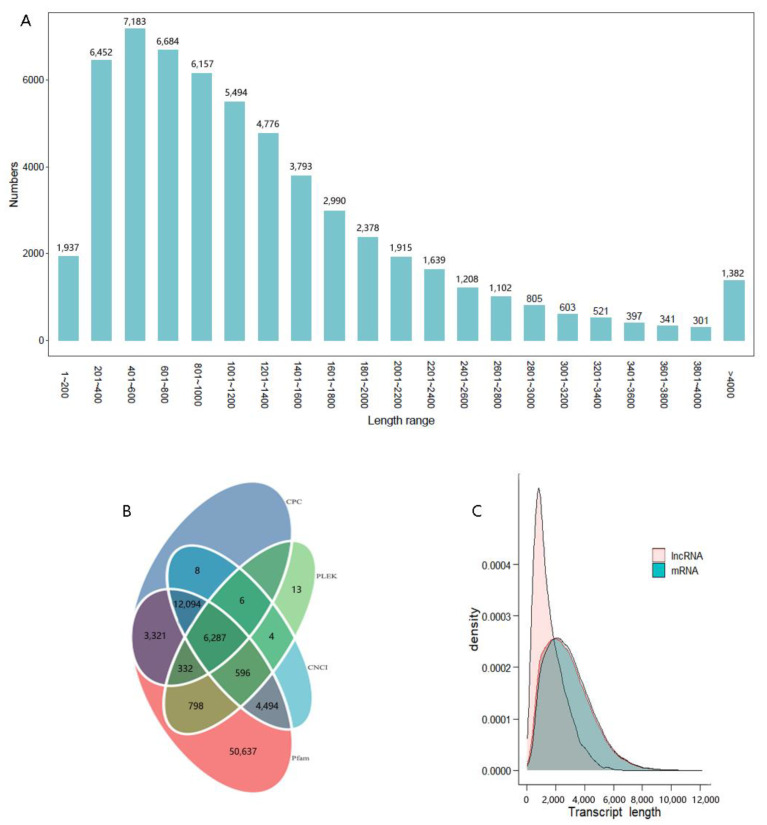
Length and number distribution of CDS from unigenes (**A**), Venn diagram of LncRNAs predicted by CPC, PLEK, CNCI, Pfam methods (**B**), and density and length distributions of LncRNAs and mRNAs (**C**) in *M. rosenbergii*.

**Figure 5 biomolecules-13-00460-f005:**
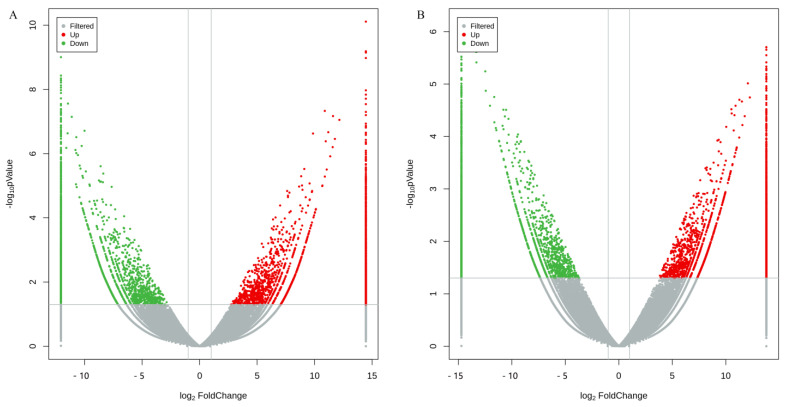
Volcanic diagrams of DEGs calculated based on raw counts from the brain (**A**) and thoracic ganglion (**B**) of *M. rosenbergii*. Statistically significant differentially expressed genes are shown in the red dots (up-regulation in male prawn) and green dots (down-regulation in female prawn), and volcanic plots were obtained by using log_2_ (fold-change) values and *p*-values.

**Figure 6 biomolecules-13-00460-f006:**
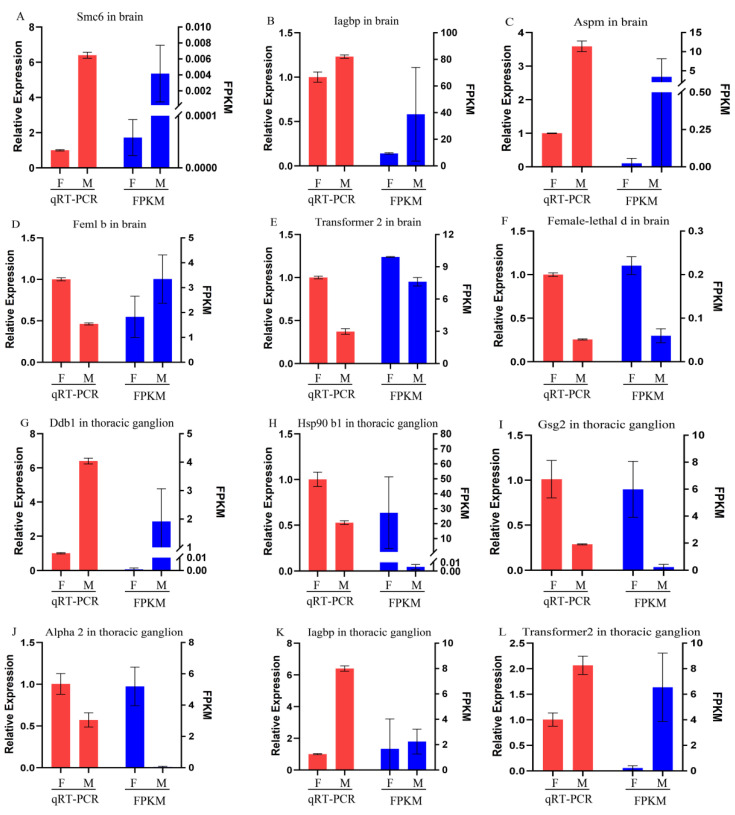
Validation of ten DEG profiles by qRT-PCR. F: female prawn, M: male prawn. The FPKM (Fragments per kilobase of transcript per million mapped reads) value was calculated by eXpress. The abbreviated genes, Smc6: Structural maintenance of chromosomes protein 6-like; Iagbp: Insulin-like androgenic gland hormone binding protein; Aspm: Abnormal spindle-like microcephaly-associated protein homolog; Ddb1: DNA damage-binding protein 1; Hsp90b1: Heat shock protein 90 beta 1; Gsg2: Ganglioside GM2 activator.

**Figure 7 biomolecules-13-00460-f007:**
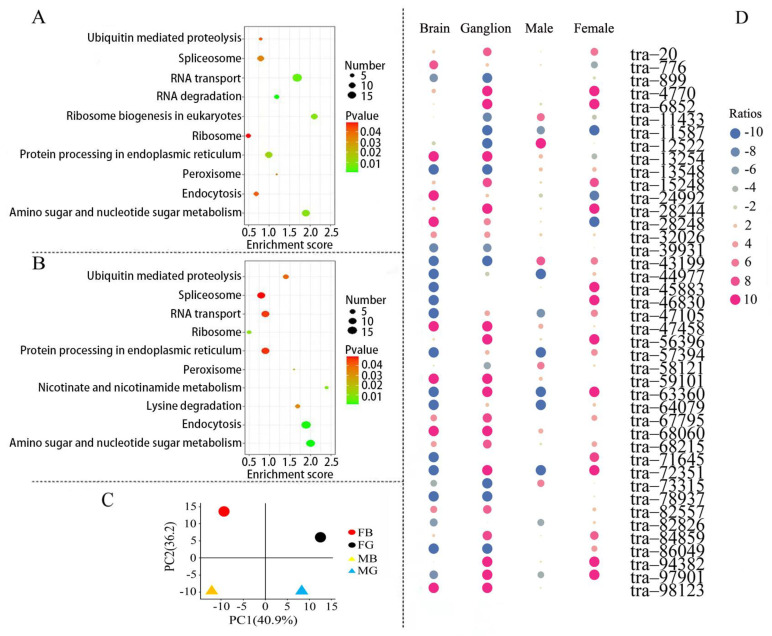
Annotation and distribution of DEGs (|log2FC| > 2, *p*-value < 0.05) in the protein processing pathway in the endoplasmic reticulum of *M. rosenbergii*. KEGG enrichment of DEGs in up-regulated expression in the brain (**A**) and up-regulated expression in the thoracic ganglion (**B**). The DEGs were separated into different principal components (**C**). The distribution is shown in (**D**) according to the tissues and sexes of *M. rosenbergii*. FB: female brain; FG: female thoracic ganglion; MB: male brain; MG: male thoracic ganglion; brain and ganglion denoted differentially expressed genes in male brain and ganglion with female as a control; male and female denoted differentially expressed genes in the male and female brain with ganglion as a control; Ratios denoted the ratio of gene expression amount in the genders and tissues, and a negative/positive number indicated down-/up-regulated expression.

**Figure 8 biomolecules-13-00460-f008:**
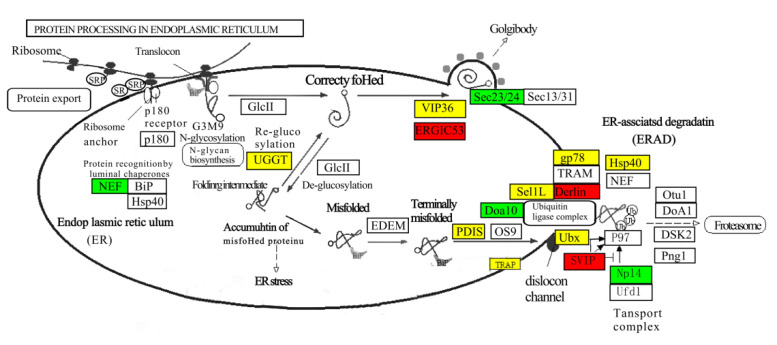
DEGs are involved in the pathway of protein processing in the endoplasmic reticulum. DEGs with up-regulated expression are colored red; DEGs with down-regulated expression are colored green, and DEGs with up/down-regulated expression are colored yellow using female prawn as a control. Genes colored blue indicate that these genes were identified from the full-length transcripts, without differential expression in the nervous system between the sexes of *M. rosenbergii*.

**Figure 9 biomolecules-13-00460-f009:**
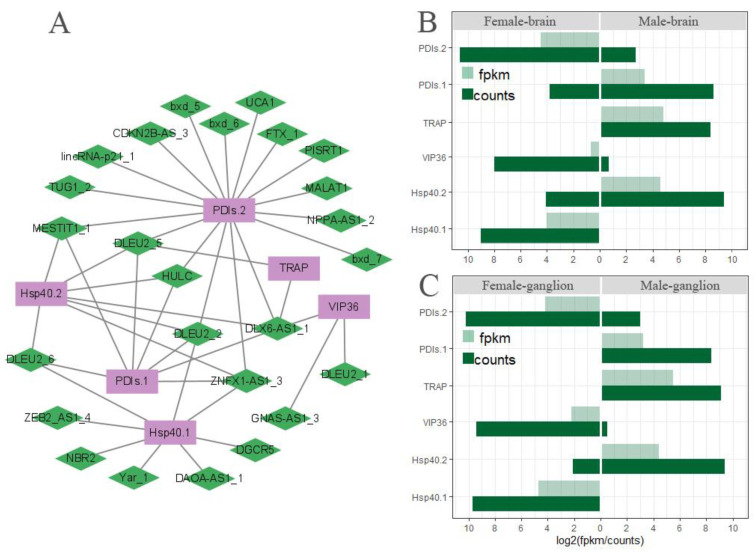
Differently expressed lncRNA and DEG co-expression networks (**A**) and absolute of log_2_(gene expression amounts) in sex-based brain (**B**) and ganglion (**C**) of *M. rosenbergii*. The purple rectangles represent genes; green diamonds represent lncRNAs. FPKM value was calculated using eXpress, and counts were calculated using bowtie2. FPKM indicates fragments per kilobase of transcript per million mapped reads.

**Table 1 biomolecules-13-00460-t001:** Sequences of primers used for gene differential expression analysis.

Unigene ID	Gene Name	Primer Sequence (5- > 3)	
		Forward	Reverse
transcript_40005	SMCP6	TGCAGTGGTTGTTGCACTTG	ACTGCACAGTAGCTGTTTGC
transcript_88064	IAGBP	AACATGGCTGACGGTTCTTC	TCCGGACGTTGATGTTCATG
transcript_16449	ASMP	ACAGATCAGTGCCACACATG	ACGGGAACAAACCATCAAGC
transcript_29898	Fem1b	AATTGCATGTGGGGCTGATG	TGTCCATGTGTGCTCCATTC
transcript_72021	transformer 2	ACATCATGGAAGCAGAGAGGAC	TGGCATCCAGAACAACTTGC
transcript_45783	female lethal d-like protein	GGTGAATTGGCCCTTCAAAGG	TTCGCCTGCCTTAATTGCTG
transcript_47685	DNA damage- binding protein 1	TTCGAGTGGACGAACGAAAG	ACGTCTTGTACTGCAGCAAG
transcript_10906	alpha-2-macroglobulin	TCAGTGAAGCAGCCTTTTGC	TGATGTCTCTGTGGCCAAAC
transcript_57773	heat shock protein 90	TCCGCAAGAACTTGGTCAAG	AGCCAACTTCTTGCGGTTAG
transcript_524	ganglioside GM2 activator	TATCGGTTTCATGTGGGTGGAG	ATGGATCAGGGCAAGGTTCG
transcript_60749	Actin	AATCGTGCGTGACATCAAGG	TCTCGTTACCGATGGTGATGAC

Abbreviated SMCP6: the structural maintenance of chromosomes protein 6-like gene; IAGBP: the insulin-like androgenic gland hormone-binding protein gene; ASMP: the abnormal spindle-like microcephaly-associated protein homolog gene.

## Data Availability

All the data generated and analyzed in this study are included in this article and additional files.

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
