# Peer review of "Full-Length Transcriptomes and Sex-Based Differentially Expressed Genes in the Brain and Ganglia of Giant River Prawn Macrobrachium rosenbergii"

_biomolecules, 2023, doi:10.3390/biom13030460_

Round 1

Reviewer 1 Report

The article presents a descriptive study of transcriptome of a previously uncharacterised prawn, Macrobrachium rosenbergii. The researchers frame the study in context of elucidating gene expression changes between fast-growing males and slow-growing females, of potential importance for the prawn farming. They also construct PacBio-based full-length transcriptome first, before using short-read Illumina sequencing for measuring gene activity, using that PacBio transcriptome as a reference sequence.

The paper does not achieve any major conclusions and the growth framing mostly disappears towards the end of the paper, owing both to some weaknesses in experimental design and (potentially) in analytic steps. However, there are no fundamental issues with the paper that I can see, although it needs major revision and clarification in many details. The data obtained by the authors will certainly be relevant to researchers working on this and similar species of economic importance.

Below are more detailed comments regarding important parts of the paper. Stylistical etc. comments are at the end.

The main issue with the paper is that it shows and reports too much of everything. Even though the paper is framed as experiment to understand / identify genes involved in much faster growth in male prawns, the results report everything and anything, almost without any filter to narrow down the answer. On its own, there is nothing wrong with reporting every finding and analysis done, but either the framing should be changed (or eveb abandoned) or the results should be selected and simplified to be more relevant to the question posed.

Now, some of these issues is derived from the fact that the experimental setup is very general and not well tailored to answering the question of the growth differences. If I understood it correctly, the Illumina sequencing was done of 3 males and 3 females, all adults. Therefore the differential expression will be only about stable differences (and large ones at this) between adult nervous system in males and females. You also have a massive confouding factor because the growth is tied to sex - how can you claim you can distinguish the two effects?

To answer the question properly, one would probably need a time series of developing prawns, or an experimental challenge where some growth hormone is injected or removed from their system, followed by RNA-seq at different timepoints, done independently on the two sexes. Or perhaps selecting unusually large females or unusually small males and running the differentially expressed genes on them.

But to characterise stable gene expression differences between the two sexes, the setup in the paper is appropriate, if weak (with only 3 samples per group it will be difficult to detect small differences in gene expression, although you do have good sequencing coverage).

My general recommendation would be to review the paper with these comments in mind and simplify the main message (with the rest safely moved to supplementary materials):

- I don't think we need Figure 1 in main body of the paper

- Similarly for Figure 2. Why are we shown so many groups in the upset plot in this figure? With such a large difference between hits to NR and the rest, perhaps the rest should be shown on a separate plot. Also, the authors should make sure the upset plot is used in the right intersect manner, as the default way it counts common elements of categories is unintuitive and misleading. I think Methods don't mention which package was used for this plot. Font size in this figure should be increased as it is illegible.

Line 106: the "three major categories" are the 3 domains of GO, all your hits by definition will always belong to these 3 categories.

- Figure 3 should be horizontal (like Figure 4) and categories should be ordered by frequency and perhaps limited to top 5? We don't know from this plot how many genes re there in each category - this should be added. As a good rule of thumb, I would remove all categories with fewer than 5 genes.

- Same comments for Figure 4.

- For Figure 5, see my comment regarding line 130.

- I would move Figure 6 to supplementary materias unless a good justification is provided for keeping it here.

- Is Figure 7 correct? It looks like a very large proportion of genes are differentially expressed. Why are there so many genes at extremes of fold change (x axis)? What fraction of genes are differentially expressed, overall and in each sex?

- Table 2 includes information of q value, which I think it not reported or mentioned elsewhere, and this value in Table 2 goes to very large numbers in several cases (>20%). It would be prudent to include the q values for the other comparisons made; q value of 47% makes me question whether that gene is indeed differentially expressed. What the other FDR values for the comparisons?

Another issue with the data in the table is the number of differentially expressed genes: I think it is not meaningful to include categories with fewer than 5 genes in there.

Finally, the table is captioned "Genes involved in growth based on GO term", but this is misleading - you cannot distinguish involvement in growth from involvement in sex differences. The caption should simply refer to GO categories of differentially expressed genes.

Line 186: Your qPCR results are not entirely consistent with RNA-seq (see eg. Feml b or Alpha 2). Again, this should be commented on but this kind of information is more suitable to supplementary methods.

Figure 9: This is a complex figure with unclear message. I can't see a clear signal and the same colours indicate different things in different panels, which makes it even harder to interpret. One expects that one figure deals with one kind of information, here each panel seems to be doing a different job. This is not Nature, we can have informative multi-panel figures :-) Please spend some time clarifying the message here.

Figure 11: I am not familiar with lncRNA analyses, so I can't comment much on this, but I am curious why do you report both FPKM values and counts here?

Many of the same comments can be repeated for Figure 10. Why is this figure important and what message does it say? The message I derive from it that a lot of coloured boxes are by the ubiquitin ligase complex, but the pattern of changes is not consistent. If the blue boxes show genes that are NOT differentially expressed, why are they highlighted? Many labels on the figure are not legible.

Throughout the manuscript, I ofted had to look for information on what comparison is being discussed: is it males vs females? brain vs ganglion? a combination of the two. These should be clarified, maybe to the point of restructuring sections of the paper dealing with one comparison at a time, or diminishing the brain vs ganglion comparisons (or moving them to supplementary materials entirely).

Methods 4.1, 4.2, 4.5

lines 328, 331, 336 and 339: 

It is not clear how many samples and of what sex were used to generate the reference (PacBio) transcriptome. In the indicated lines, the authors should clarify whether the male and female samples where mixed and in what way. Line 328 and 331 mention 6 samples in total, but line 339 mentions 3 samples.

line 367:

For the Illumina library the "total RNA was extracted from (...) 3 individuals" - is it per sex? Total? Was it 3 independent isolations followed by mixing or 1 isolation of 3 mixed samples?

In general, throughout the manuscript it was not always clear to me when the authors talk about their reference (PacBio) library and the differentially expressed genes (Illumina) library, I had to get this information from the context. I would ask the authors to read the manuscript with this in mind and clarify where needed.

Methods 4.6

line 381:

How "randomly" were differentially expressed genes selected for the qRT-PCR confirmation? What exactly was the set of genes from which you drew the 10 for qRT-PCR?

Table1:

Is is possible to have some more unequivocal gene IDs added to the table? I appreciate that there may not be a M. rosenbergii SMCP6 gene, but maybe there is a defined gene from a closely related reference species that could be used for ID?

Methods 4.7

line 397:

What does "conservative analysis of lncRNA" mean? Please clarify.

General comments

Line 22 and many others:

A concept of "unigene" is never defined in the manuscript. Please define it the first time you mention it and in the methods.

Line 23: Instead of "significant number", provide the actual number.

Line 25: Unclear what "compared to females" refers to. Please clarify.

Line 42: "The sex-linked genes on the ZW chromosomes were observed" I am not sure what is the purpose of this sentence. There are sex-linked genes on any sex chromosome, are they not? This sentence can be removed.

Line 79: What is "self-correction"? PLease explain for readers not familiar with PacBio data processing.

Line 83: "identify" should be "identity"; it is unclear to me what isoforms are and I don't understand the sentence "78,559 transcriptions were found between 1 and 10 isoforms". Are you talking about alternative splicing?

Line 103: What does it mean that it "had the highest unigene distribution"? Please clarify.

Line 110: What is an "operative catalogue"? Please explain.

Line 130: On figure 5 you plot "top 27 annotated families". This is an odd number, why 27? Please justify this choice.

Line 135: "CDSs are critically important features of mRNA transcripts that contain coding exons." Yes, this is a truism and this sentence should be removed.

Line 139: Average length of a transcript of 1557.03??? I think you need to round this number.

Line 141: "SSR has abundant polymorphisms as microsatellite markers." This is a nonsensical sentence, I think it can be removed with no harm to the rest of the paragraph.

Line 142: What is a "compound formation"? Please explain.

Line 158-159: What are "non-pollution conditions" and what does "matched in proper pairs" mean?

Line 164: What does "in proper sequence" mean?

Line 195: "In fact, male prawns showed faster growth than females, and more proteins needed to be assembled." This sentence I think is a speculation rather than a fact? Or it should be referenced. I can imagine faster growth without making more proteins, you could make them more stable or more slowly degraded, for example.

Section 2.8.

I think section 2.8. should be rewritten to make it clear what question is being answered and articulate the evidence for which lncRNA regulate what. I think the words "collaboration" and "couplings" and ERDA are either incorrectly used or not explained. Why degree = 17 is "the highest"? The sentence "The lncRNA-mRNA coupling suggested that the regulation of PDIs.2 by 230 multiple lncRNAs was likely to occur in the nervous system." sounds as if more than one tissue was being tested.

Reviewer 2 Report

The manuscript by Liu et al. represents the transcriptomic study of the central nervous system of the freshwater prawn M. rosenbergii. In general, this is an interesting study which could be important for further research of sexual dimorphism and influence of the nervous system to growth and development. Nevertheless, the manuscript cannot be published in its current form and should be substantially rewritten. First, I would recommend a professional proofreading. Second, the text sometimes looks like a pile of facts and does not provide a coherent story. This is especially pronounced inthe introduction section that does not introduce us with a problem, but rather provides a number of facts and suggestions. Of course, I understand the importance of the given study, that is clear, but the text looks like a draft, without linkages between independent suggestions and facts. Below I attach a list of comments highlighting my main points and some minor comments.
Line 20 – what kind of ganglion do you mean? From the main manuscript text it becomes clear, but it is totally obscure in the abstract. In my opinion, the abstract is an independent text, that should be self-sufficient.
Line 52-53 – that is nervous system itself, not its evolution. Evolution is a process that makes species to adopt the environmental conditions, but not to regulate body functions of a separate organisms.
Line 56 – I have never seen the term “chained neural system” concerned to crustaceans. Nervous system or ventral nerve cord or Rope-ladder-like nervous system.
Line 62: why are you writing about nanopore-based sequencing? In your study you use a different technology. So, maybe it would be more convenient to write about long-read technologies in general, not only the nanopore-based.
Results
The main question is how many biological replicates were made for differential gene expression analysis? Without replicates it is almost impossible to make strong claims and it is impossible to provide the kind of proper statistical inference on differences between groups.
Figure legends should be redone since they are sometimes too short and do not provide sufficient information.
In Figure 8, DEGs often have too large errors, which casts doubt on whether all differential expression data matches the qRT-PCR results so well. In general, this is OK, but it should have been mentioned in the text.
Matherials and methods
The experimental design is clear and simple. However, to me it stills not clear the amount of the material used for this study. How many animals/RNA samples/RNA libraries did you use? As far as I understood, the authors took 3 animals for each RNA sample (see line 331), then at line 339/40 it is written that “The total RNA from three samples was used to generate one library”. So, does it means that 3 RNA samples (made from 3 animals each) were used to make one library? How many libraries overall was made?
It is also should be noted in materials and methods section that brain and thoracic ganglia were processed separately. It is not clear from the text, however in results section the authors analyze DEGs in the brain and thoracic ganglia separately from each other.
Line 333 – what requirements do you mean? Were these tissues used for any other sequencing required for this study?
Lines 340/1 – I do not understand the importance of this step.
Line 345 – HiSeq 2500 is an Illumina Instrument? isn’t it? Something is wrong here, Looks like a copy/paste artifact.
Line 369 – what library construction kid did you use?
Line 370 – Am I right that it was only 4 cDNA libraries for Illumina sequencing? 2 samples from brains and 2 from thoracic ganglia?
Conclusions
Line 401: Do you mean “transcription factors”?

Round 2

Reviewer 1 Report

Dear authors,

thank you for the revised version of the manuscript. I believe the necessary clarifications regarding materials and methods used are now included, and I welcome the de-emphasis of the "growth" argument in favour of more appropriate "sex differences" frame.

However, I still am not satisfied with the presentation of the results, and still consider this section overwhelming and overloaded. I understand you don't agree, since you didn't implement most of my suggestions - this is fine, however I would like to have a better explanation justifying your choices. In few cases you appear to be leaving things as-is because "others do so". In few other, I think I could have articulated better what I meant.

Here is a list of my remarks.

In the abstract, please put a number in place of "significant number" of differentially expressed genes.

In line 80, a sentence "To investigate and better understand sex-linked growth dimorphism through regulation" would be better read as "To investigate and better understand sex and growth differences".

In line 86, the expression "and is responsible for growth dimorphism in the giant river prawn" is a speculation, not your actual discovery, correct? Please tone it down or remove it.

In line 128, "Hyalella azteca was the top species of unigenes distributed" - please rephrase.

My reply to your reply to point 1. (I wish there was an easier way of doing this.)

Clarification: I am not questioning the use of RNA-seq. I was trying to say that your original paper claimed to be investigating growth differences, but the actual results reported where much, much broader and not at all focused on this question (as indeed you could not answer this question with your experimental design).

My reply to your reply to point 2.

The issue with using 1-3 samples is not that you cannot find differences, but that you will not be able to detect small differences in gene expression, because your power is  very low. But you have extremely differentiated organisms to compare, so this is not a big issue.

My reply to your reply to point 4.

My major criticism of new Figure 1 stands. 70% of your hits were to 1 database, and everything else was distributed among the remaining 6, in not systematic way). Reporting this number only in text would be sufficient, the figure should at least be move to supplementary materials but is not really informative. The font size in my PDF of the revised manuscripts is too small to see the number of hits of each intersection.

What is the package this figure was generated with?

My reply to your reply to point 8.

You have now changed a seemingly arbitrary number 27 to 30, but still haven't explained where does this number come from? Why isn't it 12, or 8, or 25? Is this the number of all TF families you have found?

My reply to your reply to point 14.

My criticism regarding new Figure 8 stands and the separation of panels with dashed lines doesn't change the main point - what is the message this figure(s) convey? Just a few points to illustrate my frustration with this figure: panels A and B - why the scale p values show such large numbers if the caption suggests all the p values are < 0.05? Why x-axis on panel B goes to 5 is no value is larger than 2.6? Panel C is in my opinion unnecessary as in such extremely differentiated system you will obviously get clear separation of PCs between different sexes and tissues. Panel D - we have different numbers and colours in the legend, but no explanation what is what.

My reply to your reply to point 15.

I know that counts and FPKM are two ways of expressing transcript levels - why do you report both, particularly as they appear to be so different?

My reply to your reply to point 16.

The figure is illegible in my PDF copy of the manuscript (I hope it will look better in high resolution), but removing blue shading of the boxes would help make it better - why highlight non-differentially expressed genes?

Reviewer 2 Report

The authors addressed most my comments and suggestions. I wish you good luck with your article.

Author Response

Reply: Thank you for your bless.